

# State-of-the-art violence detection techniques in video surveillance security systems: a systematic review

Batyrkhan Omarov[1,2,3,4], Sergazi Narynov[1], Zhandos Zhumanov[1,4], Aidana Gumar[1,5] and Mariyam Khassanova[1,5]

[1] Alem Research, Almaty, Kazakhstan
[2] International University of Tourism and Hospitality, Turkistan, Kazakhstan
[3] Suleiman Demirel University, Almaty, Kazakhstan
[4] Al-Farabi Kazakh National University, Almaty, Kazakhstan
[5] Asfendiyarov Kazakh National Medical University, Almaty, Kazakhstan

## ABSTRACT

We investigate and analyze methods to violence detection in this study to completely disassemble the present condition and anticipate the emerging trends of violence discovery research. In this systematic review, we provide a comprehensive assessment of the video violence detection problems that have been described in state-of-the-art researches. This work aims to address the problems as state-of-the-art methods in video violence detection, datasets to develop and train real-time video violence detection frameworks, discuss and identify open issues in the given problem. In this study, we analyzed 80 research papers that have been selected from 154 research papers after identification, screening, and eligibility phases. As the research sources, we used five digital libraries and three high ranked computer vision conferences that were published between 2015 and 2021. We begin by briefly introducing core idea and problems of video-based violence detection; after that, we divided current techniques into three categories based on their methodologies: conventional methods, end-to-end deep learning-based methods, and machine learning-based methods. Finally, we present public datasets for testing video based violence detectionmethods' performance and compare their results. In addition, we summarize the open issues in violence detection in videoand evaluate its future tendencies.

## INTRODUCTION

Surveillance and anomaly detection have become more important as the quantity of video data has grown rapidly (*Feng, Liang & Li, 2021*). When compared to regular activity, such aberrant occurrences are uncommon. As a result, creating automated video surveillance systems for anomaly detection has become a need to reduce labor and time waste. Detecting abnormalities in films is a difficult job since the term "anomaly" is often imprecise and poorly defined (*Yang et al., 2018*). They differ greatly depending on the conditions and circumstances in which they occur. Bicycling on a standard route, for example, is a typical

Corresponding author
Batyrkhan Omarov,
Batyrkhan.Omarov2@kaznu.kz,
batyahan@gmail.com

activity, but doing so in a walk-only lane should be noted as unusual. The uneven internal occlusion is a noteworthy, yet difficult to explain characteristic of the abnormal behavior. Furthermore, owing to its large dimensionality, resolution, noise, and rapidly changing events and interactions, video data encoding and modeling are more challenging. Other difficulties include lighting changes, perspective shifts, camera movements, and so on *Yazdi & Bouwmans (2018)*.

Violence detection is one of the most crucial elements of video-based anomaly detection (*Khan et al., 2019*). The usage of video cameras to monitor individuals has become essential due to the rise in security concerns across the globe, and early detection of these violent actions may significantly minimize the dangers. A violence detection system's primary goal is to identify some kind of aberrant behavior that fits under the category of violence (*Mabrouk & Zagrouba, 2018*).

If an event's conduct differs from what one anticipates, it is considered violent. A person striking, kicking, lifting the other person, and so on are examples of such anomalies (*Shao, Cai & Wang, 2017*). An item in an unusual place, odd motion patterns such as moving in a disorganized way, abrupt motions, fallen objects are all examples of violent occurrences (*Munn et al., 2018*).

Since human monitoring of the complete video stream is impractical owing to the repetitive nature of the work and the length of time required, automated identification of violent events in real-time is required to prevent such incidents (*Tripathi, Jalal & Agrawal, 2018*).

Many scholars considered various methods to improve violence detection performance. Using a comprehensive literature review, various techniques of detecting violence from surveillance camera videos are examined and addressed in depth in this study.

The main goal of this review is to provide an in-depth, systematic overview of the techniques for detecting violence in video. Various techniques of detecting violence in video and aggressive behavior have been developed during the past decade. These techniques must be classified, analyzed, and summarized. To perform a systematic literature review, we created basic search phrases to find the most relevant studies on the detection of violent behavior accessible in five digital libraries as ScienceDirect, IEEEXplore digital library, Springer, Wiley, Scopus databases, and well-known conferences in the computer vision area as Conference on Computer Vision and Pattern Recognition (CVPR), International Conference on Computer Vision (ICCV), and European Conference on Computer Vision (ECCV).

Research highlights of this systematic review are described as follows:

- Review of state-of-the-art violence detection methods highlighting their originality, key features, and limitations.
- Study of ranking and importance of video feature descriptors for detecting violence in video.
- Exploration of datasets and evaluation criteria for violence detection in video.
- Discussion of limitations, challenges, and open issues of the video-based violence detection problem.

The remainder of this review is split into five parts. The research methodology of the current review is described in 'Research Methodology'. The fundamental idea and main concepts of violence detection in videos are discussed in 'Concepts'. The methods of violence detection in videos are explored in-depth in 'Classification of Violence Detection Techniques'. Video feature descriptors and their significance are described in 'Video Features and Descriptors'. Worldwide datasets to train the models for violence detection are discussed in 'Datasets'. Evaluation metrics that were used to test violence detection methods are described in 'Challenges to Violence Detection in Video'. Challenges and open issues in violence detection are discussed in 'Discussion'. The last section concludes the review by discussing trends, future perspectives, and open problems of violence detection.

## RESEARCH METHODOLOGY

Both qualitative and quantitative analytic techniques were integrated and used in this systematic literature review (*Ramzan et al., 2019*; *Lejmi, Khalifa & Mahjoub, 2019*). Table 1 demonstrates inclusion and exclusion criterias of the collected studies. We included two criteria as inclusion and six criteria as exclusion criteria.

Figure 1 illustrates the stages of the review as well as the number of articles that were included and eliminated. For a collection of articles, search terms and query strings were determined. The query string consists of three search terms: ''Video'' and ''Violence detection'' with the logic operator ''AND'' between them. Science Direct, IEEE Xplore, Springer, and Wiley libraries, and ''Scopus'' abstract and citation database were utilized. In addition, we asked for publications from CVPR, ICCV, and ECCV conferences. Articles published between 2015 and 2021 were considered in our study. 58 recordings were eliminated during the screening step, 16 records were eliminated during the Eligibility stage due to a lack of complete texts, duplication. As a result of the article collection, 80 research articles were included in the systematic literature review.

In our systematic literature review, we put four research questions. Table 2 demonstrates research questions and their motivations.

### Data analysis

In this subsection, we provide general analysis to the obtained results. Figure 2 demonstrates year-wise distribution of the papers that dedicated to violence detection. Figure 2A illustrates distribution of violence detection papers from January, 2016 to November, 2021. As the figure shows, the interest to the given problem increases year by year. Figure 2B presents distribution of applied methods in the selected papers. As it is illustrated in the figure, in 2016–2017 machine learning methods were popular in video violence detection problem. Moreover, we can observe decreasing of conventional methods usage, and increasing trend of deep learning based techniques.

Figure 3 demonstrates percentage of each method usage. SVM is consistently applied in the detection of violence in video occupying 24% of all methods used. Conventional methods that were used in 2015 to 2018, take about one fifth of all the applied methods. In machine learning, four algorithms frequently used in violence detection, in detail k nearest neighbors (2%), Adaptive boosting (4%), Random forest (7%), and k-means (2%).

**Table 1** Inclusion and exclusion criteria.

| I/E | Criteria | Explanation |
|---|---|---|
| **Inclusion** | Review paper | The paper proposes different types of reviews as literature review, systematic review, survey, etc. |
| | Research paper | The paper aims to solve specific research problems related to video surveillance security systems. |
| **Exclusion** | Duplicated papers | The same paper that appears multiple times |
| | Non-research papers | The paper is not a research article. It might be Editorial notes, comments, etc. |
| | Non-related papers | The topic under study goes beyond the research context of this work |
| | Non English papers | The paper is not written in English |
| | Implicitly related papers | The paper does not directly express the research focus on video surveillance security systems. |
| | Non research paper | The paper is not a research paper. It might be editorial notes, comments, etc. |

Increasing of deep learning techniques in video based violence detection can be associated with the increasing of computational performance of equipment. A total of 43% of all the applied methods use deep learning for violence detection problem. Convolutional neural networks is the most applied method for the given problem.

## CONCEPTS

The main objective of a violence detection system is to identify events in real-time so that hazardous situations may be avoided. It is, nevertheless, essential to comprehend certain key principles. Figure 4 depicts the fundamental stages of video-based violence detection methods.

### Action recognition

Action recognition is a technology that can identify human actions. Human activities are categorized into four groups based on the intricacy of the acts and the number of bodily parts engaged in the action. Gestures, actions, interactions, and group activities are the four categories (*Aggarwal & Ryoo, 2011*). A gesture is a series of motions performed with the hands, head, or other body parts to convey a certain message. A single person's actions are a compilation of numerous gestures. Interactions are a set of human activities involving at least two people. When there are two actors are involved, one should be a human and the other may be a human or an object. When there are more than two participants and one or more interacting objects, group activities involve a mix of gestures, actions, or interactions (*Aggarwal & Ryoo, 2011*).

### Violence detection

The detection of violence is a specific issue within the larger topic of action recognition. The goal of violence detection is to identify whether or not violence happens in a short amount of time automatically and efficiently. Automatic video identification of human activities has grown more essential in recent years for applications such as video surveillance,

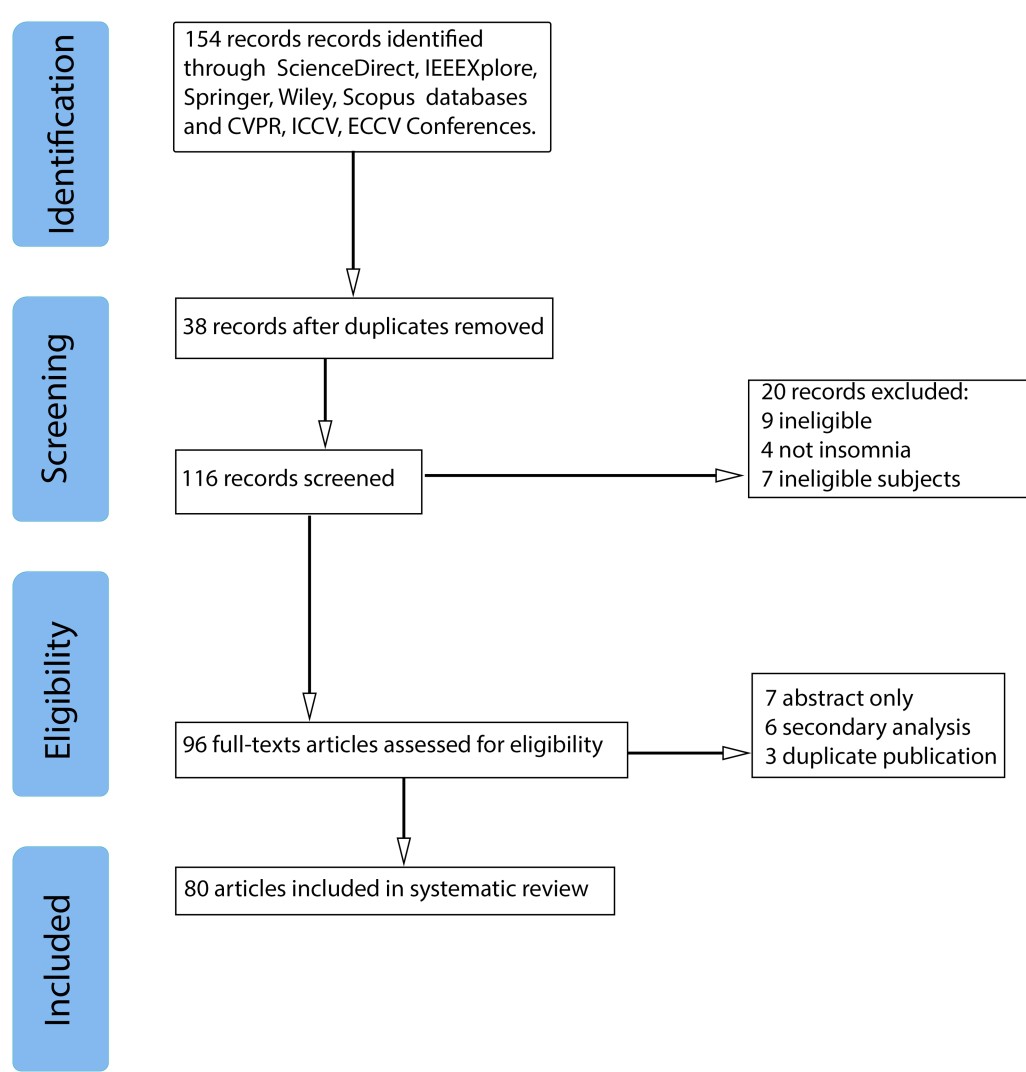

**Figure 1** Systematic literature review flowchart.

**Table 2** Research questions and their motivations.

| ID | Research Question | Motivation |
|---|---|---|
| RQ1 | What kind of video based violence detection techniques are applied in state-of-the-art researches? | Identify state-of-the-art methods and techniques in intelligent video surveillance |
| RQ2 | What kind of video features and descriptors are used in video-violence detection? | Identify commonly used and state-of-the-art features and descriptors in video violence detection |
| RQ3 | What datasets are used to train models for video-violence detection | Identify datasets commonly used in intelligent video surveillance |
| RQ4 | What challenges and open questions exist to identify violence in videos? | Identify challenges and open issues in intelligent video surveillance |

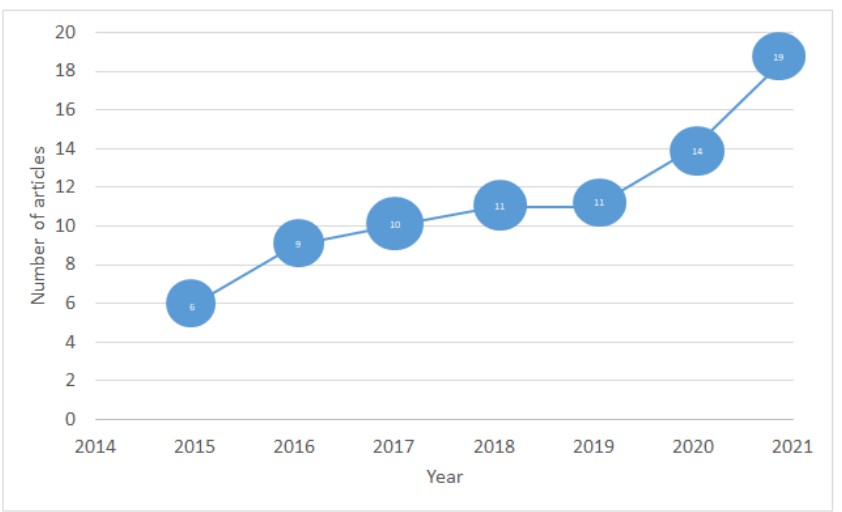

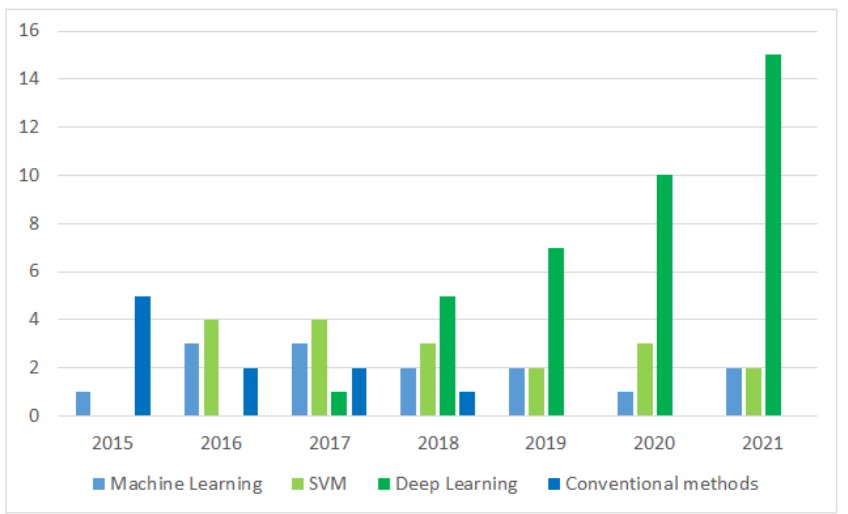

**Figure 2** Year-wise violence detection papers distribution.

human–computer interaction, and video retrieval based on content (*Poppe, 2010*; *Sun & Liu, 2013*).

The goal of violence detection is to identify whether or not violence happens automatically and effectively. In any case, detecting violence is a tough task in and of itself, since the notion of violence is subjective. Because it possesses features that distinguish it from generic acts, violence detection is a significant problem not just at the application level but also at the research level.

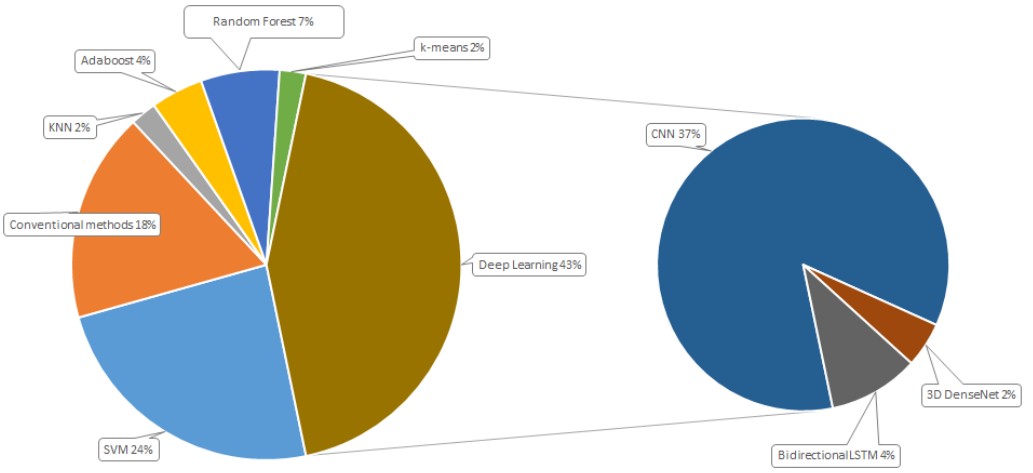

**Figure 3** Distribution of violence detection methods.

# CLASSIFICATION OF VIOLENCE DETECTION TECHNIQUES

In everyday life, violence is defined as suspicious occurrences or actions. The use of computer vision to recognize such actions in surveillance cameras has become a popular issue in the area of action recognition (*Naik & Gopalakrishna, 2017*). Scientists have presented various approaches and methods for detecting violent or unusual occurrences, citing the fast rise in crime rates as an example of the need for more efficient identification. Various methods for detecting violence have been developed in past few years. Based on the classifier employed, violence detection methods are divided into three categories: violence detection using machine learning, violence detection using SVM, and violence detection using deep learning. Because SVM and deep learning are extensively employed in computer vision, they are categorized independently. Tables explain the specifics of each technique. The techniques are given in the order in which they were developed. A methodology for detecting objects and a method for extracting features are also discussed.

## Violence detection using machine learning techniques

In this subsection, we review violence detection techniques that applied classical machine learning techniques. In Table 3, we summarize different classification techniques for violence detection in videos by indicating object detection, feature extraction, classification, the applicability of the methods for different types of scenes, and their evaluation parameters when using in different datasets. Further, we describe each technique in detail.

In the field of computer vision, action recognition has now become a relevant research area. Nevertheless, most researches have concentrated on relatively basic activities such as clapping, walking, running, and so on. The identification of particular events with immediate practical application, such as fighting or general violent conduct, has received much less attention. In certain situations, such as prisons, mental institutions, or even camera phones, video surveillance may be very helpful. A new technique for detecting violent sequences was suggested by Gracia et al. To distinguish between fighting and

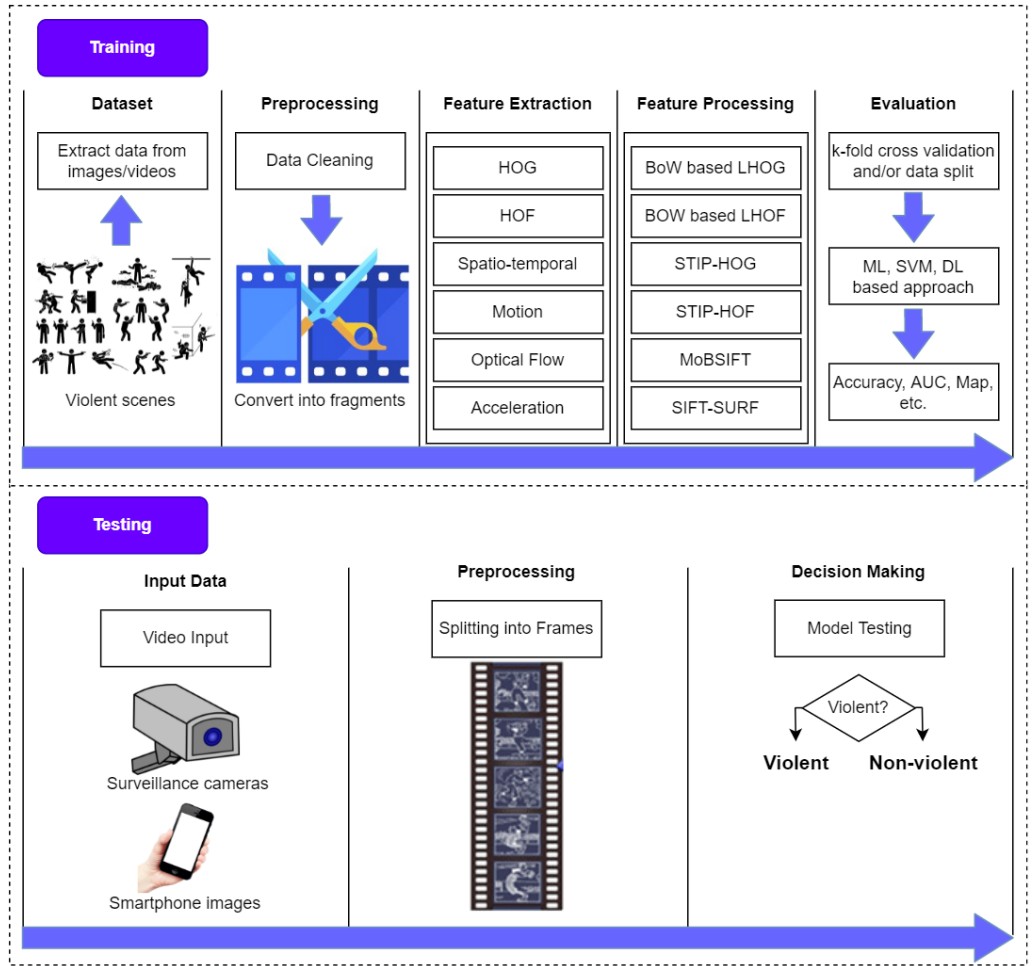

**Figure 4** **Fundamental stages of video-based violence detection.**

non-fighting episodes, features derived from motion blobs are utilized. The proposed method was assessed using three different datasets as "Movies" dataset with 200 video clips (*Bermejo et al., 2011*), the "Hockey fight" dataset that consists of 1000 video clips (*Nievas et al., 2011*), and the UCF-101 dataset of realistic action videos collected from Youtube (*Soomro, Zamir & Shah, 2012*). The proposed method was compared with other five related methods as Bag of Words (BoW) (*Wang, Wang & Fan, 2021*) using scale-invariant feature transform (MoSIFT) (*Chen & Hauptmann, 2009*) and STIP (*Ushapreethi & Lakshmi Priya, 2020*) features, Violent Flows (ViF) method (*De Souza & Pedrini, 2017*), Local Motion method (*Zhang et al., 2019*), also variant v-1 and variant v-2 methods that applied KNN, AdaBoost, and Random Forest classifiers. Although the proposed technique falls short from a perspective of performance, it has a much quicker calculation time, making it suitable for practical uses.

Automatically detecting aggressive behaviors in video surveillance situations such as train stations, schools, and mental institutions is critical. Previous detection techniques,

**Table 3  Violence detection techniques that use machine learning.**

| | | | | | |
|---|---|---|---|---|---|
| *Serrano Gracia et al. (2015)* | Motion blob acceleration measure vector method for detection of fast fighting from video | Ellipse detection method | An algorithm to find the acceleration | Spatio-temporal features use for classification | Both crowded and less crowded | Accuracy about 90% |
| *Zhou et al. (2018)* | FightNet for Violent Interaction Detection | Temporal Segment Network | Image acceleration | Softmax | Both crowded and uncrowded | 97% in Hockey, 100% in Movies dataset |
| *Ribeiro, Audigier & Pham (2016)* | RIMOC method focuses on speed and direction of an object on the base of HOF | Covariance Matrix method STV based | Spatio-temporal vector method (STV) | STV uses supervised learning | Both crowded and uncrowded | For normal situation 97% accuracy |
| *Yao et al. (2021)* | Multiview fight detection method | YOLO-V3 network | Optical flow | Random Forest | Both crowded and uncrowded | 97.66% accuracy, 97.66 F1-score |
| *Arceda et al. (2016)* | Two step detection of violent and faces in video by using ViF descriptor and normalization algorithms | Vif object recognition CUDA method and KLT face detector | Horn shrunk method for histogram | Interpolation classification | Less crowded | Lower frame rate 14% too high rate of 35% fs/s 97% |
| *Wu et al. (2020a), Wu et al. (2020b)* | HL-Net to simultaneously capture long-range relations and local distance relations | HLC approximator | CNN based model | Weak supervision | Both crowded and uncrowded scene | 78.64% |
| *Xie et al. (2016)* | SVM method for recognition based on statistical theory frames | Vector normalization method | Macro block technique for features extractions | Region motion and descripton for video classification | Crowded | 96.1% accuracy |
| *Febin, Jayasree & Joy (2020)* | A cascaded method of violence detection based on MoBSIFT and movement fltering | MoBSIFT | Motion boundary histogram | SVM, random forest, and AdaBoost | Both Crowded and uncrowded scene | 90.2% accuracy in Hockey, 91% in Movies dataset |
| *Senst et al. (2017)* | Lagrangian fields of direction and begs of word framework to recognize the violence in videos | Global compensation of object motion | Lagrangian theory and STIP method for extract motion features | Late fusion for classification | Crowded | 91% to 94% accuracy |

on the other hand, often extract descriptors surrounding spatiotemporal interesting spots or statistic characteristics in motion areas, resulting in restricted capacities to identify video-based violent activities efficiently. *Zhou et al. (2017)* present a new technique for detecting violent sequences to solve this problem. To begin, the motion areas are divided into segments based on the distribution of optical flow fields. Second, we suggest extracting two types of low-level characteristics to describe the emergence and dynamics of violent behaviors in the motion areas. The Local Histogram of Oriented Gradient (LHOG) descriptor (*Dalal & Triggs, 2005*) derived from RGB pictures and the LHOF descriptor (*Dalal, Triggs & Schmid, 2006*) extracted from optical flow images are the suggested low-level features. Finally, to remove duplicate information, the collected features are coded using the Bag of Words (BoW) model, and a specific-length vector is produced for each video clip. Finally, SVM is used to classify the video-level vectors. The suggested detection technique outperforms the prior approaches in three difficult benchmark datasets, according to experimental findings.

We chose to start at a basic level to describe what is often present in film with violent human behaviors: jerky and unstructured motion, that is due to the fact that aggressive occurrences are difficult to quantify owing to their unpredictability and sometimes need high-level interpretation. In order to capture its structure and distinguish the unstructured movements, a new problem-specific Rotation-Invariant feature modeling

MOtion Coherence (RIMOC) was suggested (*Ribeiro, Audigier & Pham, 2016*). It is based on eigenvalues calculated locally and densely from second-order statistics of Histograms of Optical Flow vectors from successive temporal instants, then embedded into a spheric Riemannian manifold. In a poorly supervised way, the proposed RIMOC feature is utilized to develop statistical models of normal coherent movements. Events with irregular mobility may be identified in space and time using a multi-scale approach combined with an inference-based method, making them ideal candidates for aggressive events. There is no special dataset available for violence and aggressive behavior detection. A big dataset is produced for this goal, which comprises of sequences from two distinct sites: an in-lab fake train and a genuine underground railway line, real train, and then four datasets are formed: fake train, real train, real train station, and real-life settings. These datasets are used in the trials, and the findings indicate that the suggested approach outperforms all state-of-the-art methods in terms of ROC per frame and false-positive rate.

Yao et al. present a multiview fight detection technique based on optical flow statistical features and random forest (*Yao et al., 2021*). This technique may provide fast and reliable information to cyber-physical monitoring systems. Motion Direction Inconsistency (MoDI) and Weighted Motion Direction Inconsistency (WMoDI), two new descriptors, are developed to enhance the performance of current techniques for films with various filming perspectives and to address misjudgment on nonfighting activities like jogging and chatting. The motion regions are first marked using the YOLO V3 method, and then the optical flow is calculated to retrieve descriptors. Finally, Random Forest is utilized to classify data using statistical descriptor features. The experiments were performed using CASIA Action Dataset (*Wang, Huang & Tan, 2007*) and the UT-Interaction Dataset (*Zhang et al., 2017*). All films of fighting, as well as 15 additional videos in five categories, were chosen from the CASIA Action Dataset. The findings demonstrated that the proposed approach improves violence detection accuracy and reduces the incidence of missing and false alarms, and it is robust against films with various shooting perspectives.

Fast face detection (*Arceda et al., 2016*) is developed to accomplish the objective of identifying faces in violent videos to improve security measures. For the initial step of violent scene identification, the authors utilized the ViF descriptor (*Keçeli & Kaya, 2017*) in conjunction with Horn-Schunck (*Keçeli & Kaya, 2017*). Then, to enhance the video quality, the non-adaptive interpolation super-resolution algorithm was used, followed by the firing of the Kanade-Lucas-Tomasi (KLT) face detector (*AlexNet, 2021*). The authors used CUDA to parallelize the super-resolution and face detection algorithms in order to achieve a very fast processing time. The Boss Dataset (*Bas, Filler & Pevný, 2011*) was utilized in the tests, as well as a violence dataset based on security camera footage. Face detection yields encouraging results in terms of area under the curve (AUC) and accuracy.

For years, computer vision researchers have been exploring how to identify violence. Prior studies, on the other hand, are either shallow, as in the categorization of short-clips and the single scenario, or undersupplied, as in the single modality and multimodality based on hand-crafted characteristics. To address this issue, the XD-Violence dataset, a large-scale and multi-scene dataset with a total length of 217 h and 4754 untrimmed films with audio signals and poor labels was proposed (*Wu et al., 2020a*; *Wu et al., 2020b*). Then, to capture

different relations among video snippets and integrate features, a neural network with three parallel branches was proposed: a holistic branch that catches long-range dependencies using similarity prior, a localized branch that captures local positional relations utilizing proximity prior, and a score branch that dynamically captures the closeness of predicted score. In addition, to fulfill the requirements of online detection, the proposed approach incorporates an approximator. Authors use the frame-level precision–recall curve (PRC) and corresponding area under the curve (average precision, AP) (*Wu et al., 2020a*; *Wu et al., 2020b*) instead of the receiver operating characteristic curve (ROC) and corresponding AUC (*Yoganand & Kavida, 2018*; *Xie et al., 2016*) because AUC typically shows an optimistic result when dealing with class-imbalanced data, whereas PRC and AP focus on positive samples (violence). The proposed approach beats other state-of-the-art algorithms in the publicly available dataset created by authors. Furthermore, numerous experimental findings indicate that multimodal (audio-visual) input and modeling connections have a beneficial impact.

Most conventional activity identification techniques' motion target detection and tracking procedures are often complex, and their applicability is limited. To solve this problem, a fast method of violent activity recognition is introduced which is based on motion vectors (*Xie et al., 2016*). First and foremost, the motion vectors were directly retrieved from compressed video segments. The motion vectors' characteristics in each frame and between frames were then evaluated, and the Region Motion Vectors (RMV) descriptor was produced. To classify the RMV to identify aggressive situations in movies, a SVM classifier with radial basis kernel function was used in the final step. In order to evaluate the proposed method, the authors created VVAR10 dataset that consists of 296 positive samples and 277 negative samples by sorting video clips from UCF sports (*Xie et al., 2016*), UCF50 (*Reddy & Shah, 2013*), HMDB51 (*Kuehne et al., 2011*) datasets. Experiments have shown that the proposed method can detect violent scenes with 96.1% accuracy in a short amount of time. That is why the proposed method can be used in embedded systems.

Most of the research in the field of action recognition has concentrated on people identification and monitoring, loitering, and other similar activities, while identification of violent acts or conflicts has received less attention. Local spatiotemporal feature extractors have been explored in previous studies; nevertheless, they come with the overhead of complicated optical flow estimates. Despite the fact that the temporal derivative is a faster alternative to optical flow, it produces a low-accuracy and scale-dependent result when used alone (*Li et al., 2018*). As a result, a cascaded approach of violence detection was suggested (*Febin, Jayasree & Joy, 2020*), based on motion boundary SIFT (MoBSIFT) and a movement filtering method. The surveillance films are examined using a movement filtering algorithm based on temporal derivatives in this approach, which avoids feature extraction for most peaceful activities. Only filtered frames may be suitable for feature extraction. Motion boundary histogram (MBH) is retrieved and merged with SIFT (*Lowe, 2004*) and histogram of optical flow feature to create MoBSIFT descriptor. The models were trained using MoBSIFT and MPEG Flow (MF) (*Kantorov & Laptev, 2014*) descriptors using AdaBoost, RF, and SVM classifiers. Because of its great tolerance to camera motions, the suggested MoBSIFT surpasses current techniques in terms of accuracy. The use of

movement filtering in conjunction with MoBSIFT has also been shown to decrease time complexity.

In computer vision, Lagrangian theory offers a comprehensive set of tools for evaluating non-local, long-term motion information. Authors propose a specialized Lagrangian method for the automatic identification of violent situations in video footage based on this theory (*Senst et al., 2017*). The authors propose a new feature based on a spatio-temporal model that utilizes appearance, background motion correction, and long-term motion information and leverages Lagrangian direction fields. They use an expanded bag-of-words method in a late-fusion way as a classification strategy on a per-video basis to guarantee suitable spatial and temporal feature sizes. Experiments were conducted in three datasets as "Hockey Fight" (*Nievas et al., 2011*), "Violence in Movies" (*Bermejo et al., 2011*), "Violent Crowd" (*Hassner, Itcher & Kliper-Gross, 2012*), and "London Metropolitan Police (London Riots 2011)" (*Cheng & Williams, 2012*) datasets. Multiple public benchmarks and non-public, real-world data from the London Metropolitan Police are used to verify the proposed system. Experimental results demonstrated that the implementation of Lagrangian theory is a useful feature in aggressive action detection and the classification efficiency rose over the state-of-the-art techniques like two-stream convolutional neural network (CNN, ConvNet), ViF, HoF+BoW with STIP, HOG+BoW with STIP, *etc.* in terms of accuracy and ROC-AUC measure.

## Violence detection techniques using SVM

The methods for detecting violence using the SVM as a classifier are described in-depth here.

A collection of SVM based violent incident recognition methods is shown in Table 4. SVM is a supervised learning method that is used to tackle classification issues. We display data on (number features) dimension space in SVM and distinguish between two groups. SVM is a popular technique in computer vision since it is robust and takes quantitative information into account. It is used to do binary classification jobs. Kernel is the foundation of SVM. Kernel is a function that transforms data into a high-dimensional space in which the issue may be solved. The lack of transparency in the findings is a significant drawback of SVM (*Auria & Moro, 2007*). SVM-based techniques for detecting violence are now described in full separately.

A new method for identifying school violence was proposed (*Ye et al., 2020*). This technique uses the KNN algorithm to identify foreground moving objects and then uses morphological processing methods to preprocess the identified targets. Then, to optimize the circumscribed rectangular frame of moving objects, a circumscribed rectangular frame integrating technique was proposed. To explain the distinctions between school violence and everyday activities, rectangular frame characteristics and optical-flow features were retrieved. To decrease the feature dimension, the Relief-F and Wrapper algorithms were applied. SVM is used as a classifier, and 5-fold cross-validation was conducted. The results show 94.4 percent precision and 89.6 percent accuracy. In order to improve recognition performance, a DT–SVM two-layer classifier is created. Authors utilized boxplots to identify certain DT layer characteristics that can differentiate between everyday

**Table 4  Violence detection techniques using SVM.**

| | | | | | |
|---|---|---|---|---|---|
| Ye et al. (2020) | A Video-Based DT–SVM School Violence Detecting Algorithm | Motion Co-occurrence Feature (MCF) | Optical flow extraction | Crowded | 97.6% |
| Zhang et al. (2016) | GMOF framework with tracking and detection module | Gaussian Mixture model | OHFO for optical flow extraction | Crowded | 82%–89% accuracy |
| Gao et al. (2016) | Violence detection using Oriented ViF | Optical Flow method | Combination of ViF and OViF descriptor | Crowded | 90% |
| Deepak, Vignesh & Chandrakala (2020) | Autocorrelation of gradients based violence detection | Motion boundary histograms | Frame based feature extraction | Crowded | 91.38% accuracy in Crowd Violence; 90.40% in Hockey dataset |
| Al-Nawashi, Al-Hazaimeh & Saraee (2017) | Framework includes preprocessing, detection of activity and image retrieval. It identifies the abnormal event and image from data-based images. | Optical flow and tempora difference for object detection CBIR method for retrieving images. | Gaussian function for video future analysis | Less crowded | 97% accuracy |
| Kamoona et al. (2019) | Sparsity-Based Naive Bayes Approach for Anomaly Detection in Real Surveillance Videos | Sparsity-Based Naive Bayes | C3D feature extraction | Both crowded and uncrowded | 64.7% F1 score; 52.1% precision; 85.3% recall in UCF dataset |
| Song, Kim & Park (2018) | SGT-based and SVM-based multi-temporal framework to detect violent events in multi-camera surveillance. | Late fusion | Multi-temporal Analysis (MtA) | Variety fight scenes from minimum two to maximum fifteen people include various movements | 78.3% (SGT-based, BEHAVE), 70.2% (SVM-based, BEHAVE), 87.2% (SGT-based, NUS–HGA), and 69.9% (SGT-based, YouTube) |
| Vashistha, Bhatnagar & Khan (2018) | An architecture to identify violence in video surveillance system using ViF and LBP | Shape and motion analysis | ViF and Local Binary Pattern (LBP) descriptors | Both crowded and non-crowded scenes | 89.1% accuracy in Hockey dataset, 88.2% accuracy in Violent-Flow dataset |

activities and physical violence. The SVM layer conducted categorization for the remaining activities. The accuracy of this DT–SVM classifier was 97.6 percent, while the precision was 97.2 percent, indicating a considerable increase.

Surveillance systems are grappling with how to identify violence. However, it has not received nearly as much attention as action recognition. Existing vision-based techniques focus mostly on detecting violence and make little attempt to pinpoint its location. To tackle this problem, Zhange et al. presented a quick and robust method for identifying and localizing violence in surveillance situations to address this issue (Zhang et al., 2016).

A Gaussian Model of Optical Flow (GMOF) is suggested for this purpose in order to extract potential violent areas, which are adaptively modeled as a departure from the usual crowd behavior seen in the picture. Following that, each video volume is subjected to violence detection by intensively sampling the potential violent areas. The authors also propose a new descriptor called the Orientation Histogram of Optical Flow (OHOF), which is input into a linear SVM for classification to differentiate violent events from peaceful ones. Experimental results on violence video datasets like "Hockey" (*Nievas et al., 2011*), "BEHAVE" (*Blunsden & Fisher, 2010*), "CAVIAR" (*Fisher, 2004*) have shown the superiority of the proposed methodology over the state-of-the-art descriptors like MoSIFT and SIFT, HOG, HOF, and Combination of HOG and HOF (HNF), in terms of detection accuracy, AUC-ROC, and processing performance, even in crowded scenes.

With more and more surveillance cameras deployed nowadays, the market need for smart violence detection is steadily increasing, despite the fact that it is still a challenging subject in the study. In order to recognize violence in videos in a realistic manner, a novel feature extraction method named Oriented VIolent Flows (OViF) was proposed by *Gao et al. (2016)*. In statistical motion orientations, the proposed method fully exploits the motion magnitude change information. The features are selected using AdaBoost, and the SVM classifier is subsequently trained on the features. Experiments are carried out on the "Hockey" and "Violent-Flow" (*Xu, Jiang & Sun, 2018*) datasets to assess the new approach's performance. The findings indicate that the suggested technique outperforms the baseline methods LTP and ViF in terms of accuracy and AUC. Furthermore, feature and multi-classifier combination methods have been shown to help improve the performance of the violence detector. The experiment results demonstrate that the combination of ViF and OViF using AdaBoost with a combination of Linear-SVM surpasses the state-of-the-art on the Violent-Flows database. The final best violence detection rates are 87.50% and 88.00% on Hockey Fight and Violent-Flows separately using ViF + OViF with Adaboost + SVM.

One of the most important stages in the development of machine learning applications is data representation. Data representation that is efficient aids in better classification across classes. Deepak et al. investigate Spatio-Temporal Autocorrelation of Gradients (STACOG) as a handmade feature for extracting violent activity characteristics from surveillance camera videos (*Deepak, Vignesh & Chandrakala, 2020*). The proposed strategy is divided into two stages: (1) Extraction of STACOG based Features features; (2) Discriminative learning of violent/non-violent behaviors using an SVM Classifier. Two well-known datasets were used to test the proposed approach. The Hockey fight dataset (*Nievas et al., 2011*) contains 1000 video clips and the Crowd Violence Dataset. The proposed "STACOG features + SVM" model shown 91.38% accuracy in violence detection overcoming state-of-the-art methods like HOF+BoW, HNF+BoF, ViF+SVM, BiLSTM, GMOF, and others.

In video processing, aggression detection is critical, and a surveillance system that can operate reliably in an academic environment has become a pressing requirement. To solve this problem, a novel framework for an automatic real-time video-based surveillance system is proposed (*Al-Nawashi, Al-Hazaimeh & Saraee, 2017*). The proposed system is divided into three phases during the development process. The first stage is preprocessing stage that includes abnormal human activity detection and content-based image retrieval (CBIR) in

the event that the system identifies unusual student behavior. In the first stage, students are registered by entering their personal data including first name, second name, birthday, course, student id card, and photos. The entered data is stored in a central database for conducting a search when abnormal actions are detected. The video is then turned into frames in the second step. Motion objects are detected using a temporal-differencing method, and motion areas are identified using the Gaussian function. Furthermore, a form model based on the OMEGA equation is employed as a filter for identified items, whether human or non-human. SVM is used to classify human behaviors into normal and abnormal categories. When a person engages in abnormal behavior, the system issues an automated warning. It also adds a method to get the identified item from the database using CBIR for object detection and verification. Finally, a software-based simulation using MATLAB is performed, and experimental findings indicate that the system performs simultaneous tracking, semantic scene learning, and abnormality detection in an academic setting without the need of humans.

*Kamoona et al. (2019)* proposes a model-based method for anomaly identification for surveillance video. There are two stages to the system. Multiple handcrafted features have been presented on this platform. Deep learning techniques have also been used to extract spatial–temporal characteristics from video data, such as C3D features (*Sultani, Chen & Shah, 2018*), as well as anomaly detection using SVM. The next phase is behavior modeling. In this phase, SVM is trained using a Bag of Visual Word (BOVW) to learn the typical behavior representation.

*Song, Kim & Park (2018)* proposes a new framework for high-level activity analysis based on late fusion and multi-independent temporal perception layers, which is based on late fusion. It is possible to manage the temporal variety of high-level activities using this approach. Multi-temporal analysis, multi-temporal perception layers, and late fusion are all part of the framework. Based on situation graph trees (SGT) and SVM, authors create two kinds of perception layers (SVMs). Through a phase of late fusion, the data from the multi-temporal perception layers are fused into an activity score. To test the proposed method, the framework is applied to the detection of violent events by visual observation. The experiments are conducted applying three well-known databases: BEHAVE (*Blunsden & Fisher, 2010*), NUS–HGA (*Zhuang et al., 2017*), and a number of YouTube videos depicting real-life situations. The tests yielded an accuracy of 70.2% (SVM), and 87.2% (SGT) in different datasets, demonstrating how the proposed multi-temporal technique outperforms single-temporal approaches. *Vashistha, Bhatnagar & Khan (2018)* utilized Linear SVM to categorize incoming video as violent or non-violent, extracting important characteristics like centroid, direction, velocity, and dimensions. Their approach took into account two feature vectors, *i.e.*, ViF and the Local Binary Pattern (LBP). Because calculating LBP or ViF individually it takes less time than combining these feature vectors, their study found that combining LBP and ViF did not offer substantial direction for future development.

## Violence detection techniques using deep learning

The methods for detecting violence that utilizes deep learning algorithms in the suggested frameworks are described in-depth here. Convolutional Neural Netowks (CNN) (*Zhang et al., 2019*) and its imrovemebts are widely used in violence detection in video. Table 5 shows a collection of recognition techniques that are based on deep learning. Neural networks are the foundation of deep learning. Using additional convolutional layers, the method is utilized to categorize the violent recognition based on the data set and retrieved features. Now, techniques for detecting violence that utilize deep learning algorithms are discussed in depth individually.

While most studies have focused on the issue of action recognition, fighting detection has received much less attention. This skill may be very valuable. To build complicated handmade characteristics from inputs, most techniques require on domain expertise. Deep learning methods, on the other hand, may operate directly on raw inputs and extract necessary features automatically. As a result, Ding et al. created a new 3D ConvNets approach for video violence detection, which does not need any previous information (*Ding et al., 2014*). The convolution on the collection of video frames is computed using a 3D CNN, and therefore motion information is retrieved from the input data. The back-propagation technique is used to obtain gradients and the model has been trained to apply supervised learning. Experimental validation was carried out in the context of the "Hockey fights" (*Nievas et al., 2011*) dataset to assess the approach. The findings indicate that the approach outperforms manual features in terms of performance.

Campus violence is a worldwide social phenomenon that is the most dangerous sort of school bullying occurrence. There are various possible strategies to identify campus violence as AI and remote monitoring capabilities advance, such as video-based techniques. *Ye et al. (2021)* combine visual and audio data for campus violence detection. Role-playing is used for campus violence data collection, and 4096-dimension feature vectors are extracted from every 16 frames of video frames. For feature extraction and classification, the 3D CNN is used, and overall precision of 92.00 percent is attained. Three speech emotion datasets are used to extract mel-frequency cepstral coefficients (MFCCs) as acoustic features: CASIA dataset (*Wang, Huang & Tan, 2007*) that has 960 samples, Finnish emotional dataset (*Vaaras et al., 2021*) that consists of 132 samples, and Chinese emotional dataset (*Poria et al., 2018*) that has 370 samples.

An enhanced Dempster–Shafer (D–S) algorithm is proposed to handle the problem of evidence dispute. As a result, recognition accuracy reached 97%.

To address the issue of large-scale visual place identification, the NetVLAD architecture is presented, where the goal is to rapidly and correctly identify the location of a supplied query image (*Arandjelovic et al., 2016*). NetVLAD is a CNN-based approach for weakly supervised place recognition. In this work, three major contributions are presented. First, for the location recognition problem, CNN architecture is created that can be trained in a direct end-to-end way. The central feature of this approach, NetVLAD, is a novel generalized VLAD layer inspired by the widely used picture format "Vector of Locally Aggregated Descriptors (VLAD)". The layer may be easily integrated into any CNN model and trained using backpropagation. The second contribution is the construction of a

**Table 5  Violence detection using deep learning techniques.**

| | | | | | |
|---|---|---|---|---|---|
| *Ding et al. (2014)* | Violence Detection using 3D CNN | 3D convolution is used to get spatial information | Backpropagation method | Crowded | 91% accuracy |
| *Arandjelovic et al. (2016)* | Deep architecture for place recognition | VGG VLAD method for image retrieval | Backpropagation method for feature extraction | Crowded | 87%–96% accuracy |
| *Fenil et al. (2019)* | Framework for football stadium comprising of big data analysis and deep learning through bidirectional LSTM | Bidirectional LSTM | HOG, SVM | Crowded | 94.5% accuracy |
| *Mu, Cao & Jin (2016)* | Violent scene detection using CNN and deep audio features | MFB | CNN | Crowded | Approximately 90% accuracy |
| *Mohtavipour, Saeidi & Arabsorkhi (2021)* | A multi-stream CNN using handcrafted features | A deep violence detection framework based on the specific features (speed of vmovement, and representative image) derived from handcrafted methods. | CNN | Both crowded and uncrowded | |
| *Sudhakaran & Lanz (2017)* | Detect violent videos using ConvLSTM | CNN along with the ConvLSTM | CNN | Crowded | Approximately 97% |
| *Naik & Gopalakrishna (2021)* | Deep violence detection framework based on the specific features derived from handcrafted methods | Discriminative feature with a novel differential motion energy image | CNN | Both crowded and uncrowded | |
| *Meng, Yuan & Li (2017)* | Detecting Human Violent Behavior by integrating trajectory and Deep CNN | Deep CNN | Optical flow method | Crowded | 98% accuracy |
| *Rendón-Segador et al. (2021)* | ViolenceNet: Dense Multi-Head Self-Attention with Bidirectional Convolutional LSTM | 3D DenseNet | Optical flow method | Crowded | 95.6%–100% accuracy |
| *Xia et al. (2018)* | Violence detection method based on a bi-channels CNN and the SVM. | Linear SVM | Bi-channels CNN | Both crowded and uncrowded scenes | $95.90 \pm 3.53$ accuracy in Hockey fight, $93.25 \pm 2.34$ accuracy in Violence crowd |
| *Meng et al. (2020)* | Trajectory-Pooled Deep Convolutional Networks | ConvNet model which contains 17 convolutionpool-norm layers and two fully connected layers | Deep ConvNet model | Both crowded and uncrowded | 92.5% accuracy in Crowd Violence, 98.6% in Hockey Fight dataset |
| *Ullah et al. (2019)* | Violence Detection using Spatiotemporal Features | Pre-train Mobile Net CNN model | 3D CNN | Crowded | Approximately 97% accuracy |

training method based on a novel weakly supervised ranking loss, to learn architectural parameters in an end-to-end way using Google Street View Time Machine pictures showing the same locations over time. Finally, using the Pittsburgh (*Torii et al., 2013*) and Tokyo 24/7 (*Torii et al., 2015*) datasets, the authors demonstrated that the proposed architecture outperforms non-learned image representations and off-the-shelf CNN descriptors on two difficult place recognition benchmarks, and outperforms current state-of-the-art image representations on standard image retrieval benchmarks.

A real-time violence detection system is presented (*Fenil et al., 2019*), which analyzes large amounts of streaming data and recognizes aggression using a human intelligence simulation. The system's input is a massive quantity of real-time video that feeds from various sources, which are analyzed using the Spark framework. The frames are split and the characteristics of individual frames are retrieved using the HOG function in the Spark framework. The frames are then labeled based on characteristics such as the violence model, human component model, and negative model, which are trained using the BDLSTM network for violent scene detection. The data may be accessed in both directions *via* the bidirectional LSTM. As a result, the output is produced in the context of both past and future data. The violent interaction dataset (VID) is used to train the network, which contains 2314 movies with 1077 fights and 1237 no-fights. The authors also generated a dataset of 410 video episodes with neutral scenes and 409 video episodes with violence. The accuracy of 94.5% in detecting violent behavior validates the model's performance and demonstrates the system's durability.

*Mu, Cao & Jin (2016)* are presented a violent scene identification method based on acoustic data from video. CNN in two ways: as a classifier and as a deep acoustic feature extractor. To begin, the 40-dimensional Mel Filter-Bank (MFB) is used as the CNN's input feature. The video is then divided into little pieces. To investigate the local features, MFB features are split into three feature maps. Then CNN is utilized to represent features. The CNN-based features are applied to construct SVM classifiers. Then the violent scene detection process is applied to each frame of video. After that, detection is generated by applying maximum or minimum pooling to the segment level. Experiments are conducted using the MediaEval dataset (*Demarty et al., 2014*), and the findings indicate that the proposed approach outperforms the fundamental techniques in terms of average precision: audio alone, visual solely, and audio learned fusion and visual.

A new deep violence detection approach based on handcrafted techniques' distinctive characteristics was presented (*Mohtavipour, Saeidi & Arabsorkhi, 2021*). These characteristics are linked to appearance, movement speed, and representative images, and they are supplied to a CNN as spatial, temporal, and spatiotemporal streams. With each frame, the spatial stream teaches the neural network how to recognize patterns in the surroundings. With a modified differential magnitude of optical flow, the temporal stream included three successive frames to learn motion patterns of aggressive behavior. Furthermore, the authors developed a discriminative feature with a new differential motion energy picture in the spatio-temporal stream to make violent behaviors more understandable. By combining the findings of several streams, this method includes many elements of aggressive conduct. The proposed CNN network was trained using three datasets: Hockey (*Nievas et al., 2011*), Movie (*Bermejo et al., 2011*), and ViF (*Rota et al., 2015*). The proposed method beat state-of-the-art approaches in terms of accuracy and processing time.

*Sudhakaran & Lanz (2017)* proposed a deep neural network for detecting violent scenes in videos. To extract frame-level characteristics from a video, a CNN is applied. The frame-level characteristics are then accumulated using LSTM that uses a convolutional gate. The CNN, in combination with the ConvLSTM, can capture localized spatio-temporal

characteristics, allowing for the analysis of local motion in the video. The paper also proposed feeding the model neighboring frame differences as input, pushing it to encode the changes in the video. In terms of recognition accuracy, the presented feature extraction process is tested on three common benchmark datasets as "Hockey" (*Nievas et al., 2011*), "Movies" (*Bermejo et al., 2011*), and "Violent-Flows" (*Xu, Jiang & Sun, 2018*). Findings were compared to those produced using state-of-the-art methods. It was discovered that the suggested method had a promising capacity for identifying violent films prevailing state-of-the-art methods as three streams + LSTM, ViF, and ViF+OViF.

To identify violent behaviors of a single person, an ensemble model of the Mask RCNN and LSTM was proposed (*Naik & Gopalakrishna, 2021*). Initially, human key points and masks were extracted, and then temporal information was captured. Experiments have been performed in datasets as Weizmann (*Blank et al., 2005*), KTH (*Schuldt, Laptev & Caputo, 2004*), and own Dataset respectively. The results demonstrated that the proposed model outperforms individual models showing a violence detection accuracy rate of 93.4% in its best result. The proposed approach is more relevant to the industry, which is beneficial to society in terms of security.

Typical approaches depend on hand-crafted characteristics, which may be insufficiently discriminative for the job of recognizing violent actions. Inspired by the good performance of deep learning-based approaches, propose a novel method for human violent behavior detection in videos by incorporating trajectory and deep CNN, that includes the advantage of hand-crafted features and deep-learned features (*Meng, Yuan & Li, 2017*). To assess the proposed method, tests on two distinct violence datasets are performed: "Hockey Fights" (*Nievas et al., 2011*) and "Crowd Violence" (*Song et al., 2019*) dataset. On these datasets, the findings show that the proposed approach outperforms state-of-the-art methods like HOG, HOF, ViF, and others.

*Rendón-Segador et al. (2021)* present a new approach for determining whether a video has a violent scene or not, based on an adapted 3D DenseNet, for a multi-head self-attention layer, and a bidirectional ConvLSTM module that enables encoding relevant spatio-temporal features. In addition, an ablation analysis of the input frames is carried out, comparing dense optical flow and neighboring frames removal, as well as the effect of the attention layer, revealing that combining optical flow and the attention mechanism enhances findings by up to 4.4 percent. The experiments were performed using four datasets, exceeding state-of-the-art methods, reducing the number of network parameters needed (4.5 million), and increasing its efficiency in test accuracy (from 95.6 percent on the most complex dataset to 100 percent on the simplest), and inference time (from 95.6 percent on the most complex dataset to 100 percent on the simplest dataset) (less than 0.3 s for the longest clips).

Human action recognition has become a major research topic in computer vision. Tasks like violent conduct or fights have been researched less, but they may be helpful in a variety of surveillance video situations such as jails, mental hospitals, or even on a personal mobile phone. Their broad applicability piques interest in developing violence or fight detectors. The main feature of the detectors is efficiency, which implies that these methods should be computationally quick. Although handcrafted spatio-temporal characteristics attain

excellent accuracy for both appearance and motion, extraction of certain features remains prohibitive for practical uses. For the first time, the deep learning paradigm is applied to a job using a 3D CNN that accepts the whole video stream as input. However, motion characteristics are critical for this job, and utilizing full video as input causes noise and duplication in the learning process. A hybrid feature "handcrafted/learned" framework was developed for this purpose (*Serrano et al., 2018*). The technique attempts to get an illustrative picture from the video sequence used as an input for feature extraction, using Hough forest as a classifier. 2D CNN is then utilized to categorize that picture and determine the sequence's conclusion. Experiments are carried out on three violence detection datasets as "Hockey" (*Nievas et al., 2011*), "Movie" (*Bermejo et al., 2011*), and "Behavior" (*Zhou et al., 2018*). The findings show that the suggested approach outperforms the various handmade and deep learning methods in terms of accuracies and standard deviations.

Two-stream CNN architecture, as well as an SVM classifier, is proposed (*Xia et al., 2018*). Feature extraction, training, and label fusion are the three phases of the method. Each stream CNN employs an Imagenet VGG-f architecture that has been pre-trained. The first stream collects visual information from successive frame differences, whereas the second stream extracts motion data. Then, using sight and motion information, two SVM classifiers are trained. Finally, a label fusion technique is used to get the detection result. The primary benefit of this technique is that it takes very little time to process. However, since this technique can not identify aggressive behaviors amongst individuals at close range, it is difficult to detect violence in large groups. Accattoli used two-stream CNNs in a similar way (*Accattoli et al., 2020*). To capture long temporal information, they suggest combining CNNs with better trajectories. To extract geographical and temporal information, they utilize two VGG-19 networks. Video frames are used to extract spatial information, while dense optical flow pictures are used to retrieve temporal information.

In smart cities, schools, hospitals, and other surveillance domains, an improved security system is required for the identification of violent or aberrant actions in order to prevent any casualties that may result in social, economic, or environmental harm. For this purpose, a three-staged end-to-end deep learning violence detection system is presented (*Ullah et al., 2019*). To minimize and overcome the excessive processing of use-less frames, people are first identified in the surveillance video stream using a lightweight CNN model. Second, a 16-frame sequence containing identified people is sent to 3D CNN, which extracts the spatiotemporal characteristics of the sequences and feeds them to the Softmax classifier. The authors also used open visual inference and neural networks optimization tools created by Intel to optimize the 3D CNN model, which transforms the training model into intermediate representation and modifies it for optimum execution at the end platform for the ultimate prediction of violent behavior. When violent behavior is detected, an alarm is sent to the closest police station or security agency so that immediate preventative measures may be taken. The datasets "Violent Crowd" (*Hassner, Itcher & Kliper-Gross, 2012*), "Hockey" (*Nievas et al., 2011*), and "Violence in Movies" (*Bermejo et al., 2011*) are used in the experiments. The experimental findings show that the proposed approach outperforms state-of-the-art algorithms such as ViF, AdaBoost, SVM, Hough Forest, and 2D CNN, sHOT, and others in terms of accuracy, precision, recall, and AUC.

# VIDEO FEATURES AND DESCRIPTORS

This section goes through the feature descriptors that violence detection papers utilized in their research as well as other recent state-of-the-art descriptors.

The fundamental components for detecting activity from the video are video features. The dataset and characteristics collected from video to evaluate the pattern of activity have a direct impact on the methodology's accuracy. For example, in combat situations, the movement of various objects increases faster. The movement of objects in a typical setting is normal and not too rapid. The direction of item movement in relation to time and space is also utilized to investigate unusual occurrences. Table 6 lists all of the features that were utilized in the research.

A number of scholars, such as *Lejmi, Khalifa & Mahjoub (2019)*, have worked hard to identify fights and physical violence. Previous studies (*Aggarwal & Ryoo, 2011*; *Poppe, 2010*) used blood or explosions as signals of violence, but these cues are seldom alarming. One study recently developed a feature that offers strong multimodal audio and visual signals by first combining the audio and visual characteristics and then exposing the combined multi-modal patterns statistically (*Sun & Liu, 2013*). Multiple kernel learning is used to increase the multimodality of movies by combining visual and audio data (*Naik & Gopalakrishna, 2017*). Audio-based techniques, on the other hand, are always constrained in real life due to the lack of an audio channel.

The problem of detecting violent interactions is basically one of action recognition. The objective is to extract characteristics that may describe the sequences throughout the battle using computer vision technology. Handcrafted features and learning features are the two types of features available.

Hand-crafted features. Human-designed features are referred to as hand-crafted features. In action recognition, Space-Time Interest Points (STIPs) (*Serrano Gracia et al., 2015*) and Improved Dense Trajectories (iDTs) (*Bermejo et al., 2011*) are often employed. *Deniz et al. (2014)* proposed a new approach for detecting violent sequences that utilizes severe acceleration patterns as the primary characteristic and applies the Radon transform on the power spectrum of successive frames to identify violent sequences (*Nievas et al., 2011*). *Yang et al. (2018)* presented additional characteristics derived from motion blobs between successive frames to identify combat and non-fight sequences recently. A robust and understandable method based on motion statistical characteristics from optical flow pictures was presented in similar works (*Yazdi & Bouwmans, 2018*; *Soomro, Zamir & Shah, 2012*). *Zhang et al. (2019)* use a GMOF to identify potential violence areas and a linear SVM with OHOF input vectors to discover fight regions. This kind of approach based on hand-crafted features is simple and effective for a small-scale dataset, but when used to a large dataset, its deficiencies are exposed, resulting in slow training times, large memory consumption, and inefficient execution.

Learning features. Deep neural networks learn features, which are referred to as learning features. Physical violence detection based on deep learning has made significant progress because of the increase in computing power brought on by GPUs and the gathering of large-scale training sets. Two-stream ConvNets were created (*Wang, Wang & Fan, 2021*)

**Table 6  Video features were used in the selected studies.**

| | |
|---|---|
| *Mabrouk & Zagrouba (2018)* | Motion, space and time |
| *Lejmi, Khalifa & Mahjoub (2019)* | Motion blobs, Edges and corner of image |
| *Serrano et al. (2015)* | Motion blobs |
| *Zhou et al. (2017)* | Optical flow, motion and moving bob |
| *Ribeiro, Audigier & Pham (2016)* | Motion, direction and speed |
| *Yao et al. (2021)* | MoDI and WMoDI, motion regions marking |
| *Arceda et al. (2016)* | Optical flow, Magnitude |
| *Wu et al. (2020a)*, *Wu et al. (2020b)* | Optical flow and audio features |
| *Xie et al. (2016)* | Motion vector and direction |
| *Febin, Jayasree & Joy (2020)* | Optical flow, MBH, movement filtering? |
| *Senst et al. (2017)* | Spatial, temporal and motion |
| *Ye et al. (2018)* | Rectangular frame and optical-flow |
| *Zhang et al. (2016)* | Spatiotemporal and motion |
| *Deepak, Vignesh & Chandrakala (2020)* | Spatio-Temporal Auto-Correlation of Gradients |
| *Al-Nawashi, Al-Hazaimeh & Saraee (2017)* | Motion region and optical flow |
| *Kamoona et al. (2019)* | Multiple handcrafted features |
| *Sultani, Chen & Shah (2018)* | Spatial–temporal, C3D |
| *Song, Kim & Park (2018)* | Movement, direction and speed |
| *Vashistha, Bhatnagar & Khan (2018)* | Speed, direction, centroid and dimensions |
| *Arandjelovic et al. (2016)* | Spatiotemporal features |
| *Torii et al. (2015)* | Spatio-temporal |
| *Mohtavipour, Saeidi & Arabsorkhi (2021)* | Spatial, temporal, and spatiotemporal streams |
| *Rendón-Segador et al. (2021)* | Spatio-temporal features |
| *Serrano et al. (2018)* | Apperancde, motion, optical flow |
| *Xia et al. (2018)* | Direction and motion information |
| *Accattoli et al. (2020)* | hand-crafted and trajectory-deep features |
| *Ullah et al. (2019)* | Spatiotemporal features |
| *Fu et al. (2015)* | Motion, acceleration and magnitude |
| *Deniz et al. (2014)* | Spatiotemporal, acceleration and motion |
| *Ye et al. (2018)* | Time-domain and frequency-domain |
| *Mabrouk & Zagrouba (2017)* | STIP, optical flow |

and comprise spatial and temporal nets that use the ImageNet dataset (*Chen & Hauptmann, 2009*) for pre-training and optical flow to explicitly capture motion information. Tran et al. used 3D ConvNets (*De Souza & Pedrini, 2017*) trained on a large-scale supervised dataset to learn both appearance and motion characteristics. *Zhang et al. (2019)* recently used long-range temporal structure (LTC) neural networks to train a movie and found that LTC-CNN models with increasing temporal extents enhanced action identification accuracy. However, because of the computational complexity, these techniques are restricted to a video frame rate of no more than 120 frames. The temporal segment network (*Shao, Cai & Wang, 2017*) used a sparse temporal sampling approach with video-level supervision to learn valid information from the whole action video, attaining state-of-the-art performance on the two difficult datasets HMDB51 (69.4 percent) and UCF101 (94.2 percent).

### Histogram of oriented gradients (HOG)

HOGs are a feature descriptor for object identification and localization that can compete with DNN's performance (*Dalal & Triggs, 2005*). The gradient direction distribution is utilized as a feature in HOG. Because the brightness of corners and edges vary greatly, calculating the gradient together with the directions may assist in the detection of this knowledge from the images.

### Histogram of optical flow (HOF)

A pattern of apparent motion of objects, surfaces, and edges is produced as a result of the relative motion between an observer and a scene. This process is called Optical Flow. The histogram of oriented optical flow (HOF) (*Dalal, Triggs & Schmid, 2006*) is an optical flow characteristic that depicts the series of events at each point in time. It is scale-invariant and unaffected by motion direction.

### SPACE –time interest points

Laptev and Lindeberg and Laptev proposed the space–temporal interest point detector by expanding the Harris detector. A second-moment matrix is generated for each spatiotemporal interest point after removing points with high gradient magnitude using a 3D Harris corner detector (*Laptev & Lindeberg, 2004*; *Laptev, 2005*). This descriptor's characteristics are used to describe the spatiotemporal, local motion, and appearance information in volumes.

Space-Time Interest Points (STIP) is a space–time extension of the Harris corner detection operator. The measured interest spots have a significant degree of intensity fluctuation in space and non-constant mobility in time. These important sites may be found on a variety of geographical and temporal scales. Then, for 3D video patches in the vicinity of the recognized STIPs, HOG, HOF, and a combination of HOG and HOF called HNF feature vectors are retrieved. These characteristics may be utilized to recognize motion events with high accuracy, and they are resistant to changes in pattern size, frequency, and velocity.

### MoSIFT

MoSIFT (*Chen & Hauptmann, 2009*) is an extension of the popular SIFT (*Lowe, 2004*) image descriptor for video. The standard SIFT extracts histograms of oriented gradients in the image. The 256-dimensional MoSIFT descriptor consists of two portions: a standard SIFT image descriptor and an analogous HOF, which represents local motion. These descriptors are extracted only from regions of the image with sufficient motion. The MoSIFT descriptor has shown better performance in recognition accuracy than other state-of-the-art descriptors (*Chen & Hauptmann, 2009*) but the approach is significantly more computationally expensive than STIP.

### Violence flow descriptor

The violence flow, which utilizes the frequencies of discrete values in a vectorized form, is an essential feature descriptor. This is different from other descriptors in that instead of assessing magnitudes of temporal information, the magnitudes are compared for each,

resulting in much more meaningful measurements in terms of the previous frame (*Zhang et al., 2019*). Instead of looking at local appearances, the similarities between flow-magnitudes in terms of time are investigated.

### Bag-of-Words (BoW)

The Bag-of-Words (BoW) method, which originated in the text retrieval community (*Laptev, 2005*), has lately gained popularity for a picture (*Lewis, 1998*) and video comprehension (*Csurka et al., 2004*). Each video sequence is represented as a histogram over a collection of visual words in this method, which results in a fixed-dimensional encoding that can be analyzed with a conventional classifier. The cluster centers produced *via* k-means clustering across a large collection of sample low-level descriptors are usually described as the lexicon of visual words in a learning phase (*Lopes et al., 2010*).

### Motion boundary histograms

By measuring derivatives independently for the horizontal and vertical components of the optical flow, Dalal et al. developed the MBH descriptor (*Dalal, Triggs & Schmid, 2006*) for human detection. The relative motion between pixels is encoded by the descriptor. Because MBH depicts the gradient of the optical flow, information regarding changes in the flow field (*i.e.,* motion boundaries) is preserved while locally constant camera motion is eliminated. MBH is more resistant to camera motion than optical flow, making it better at action detection.

### Vector of locally aggregated descriptors

*Soltanian, Amini & Ghaemmaghami (2019)* presented a state-of-the-art VLAD descriptor. VLAD varies from the BoW image descriptor in that it records the difference between the cluster center and the number of SIFTs allocated to the cluster, rather than the number of SIFTs assigned to the cluster. It inherits parts of the original SIFT descriptor's invariances, such as in-plane rotational invariance, and is tolerable to additional changes like picture scaling and clipping. VLAD retrieval systems typically do not utilize the original local descriptors, which is another variation from the conventional BoW method. These are employed in BoW systems for spatial verification and reranking (*Jégou et al., 2010*; *Jegou, Douze & Schmid, 2008*), but for extremely big picture datasets, they need too much storage to be kept in memory on a single machine. VLAD is comparable to the previous Fisher vectors (*Philbin et al., 2007*) in that they both store features of the SIFT distribution given to a cluster center.

VLAD is made up of areas taken from an image using an affine invariant detector and characterized with the 128-D SIFT descriptor. The nearest cluster of a vocabulary of size k is then given to each description (where k is typically 64 or 256 so that clusters are quite coarse). The residuals (vector discrepancies between descriptors and cluster centers) are collected for each of the k clusters, and the k 128-D sums of residuals are concatenated into a single k 128-D descriptor (*Philbin et al., 2007*). VLAD is comparable to other residual descriptors such as Fisher vectors (*Arandjelovic & Zisserman, 2013*) and super-vector coding (*Perronnin & Dance, 2007*).

## MoBSIFT

The MoBSIFT descriptor is a mixture of the MoSIFT and MBH descriptors. The two main stages of the MoBSIFT method are interest point identification and feature description. The video is converted into a few interest points once interest points are detected, and a feature description is done locally around these interest points.

By combining the MBH (*Dalal, Triggs & Schmid, 2006*) with the movement filtering technique, the MoSIFT descriptor (*Chen & Hauptmann, 2009*) is enhanced in both accuracy and complexity. Camera motion is a significant issue for any system since motion data is considered an essential signal in action detection. MBH is thought to be a useful feature for avoiding the effects of camera motion. It is suggested that movement filtering be used to decrease complexity by excluding most nonviolent movies from complicated feature extraction (*Zhou et al., 2010*).

## DATASETS

The real-world datasets that are utilized to evaluate the proposed violence detection methods are described in this section. The specifics of all datasets linked to the violence are summarized in Table 7.

## EVALUATION PARAMETERS

In this section, we describe evaluation parameters that are used to test the performance of physical violence detection systems.

To evaluate the effectiveness of classification algorithms, the following indicators are usually used: accuracy, completeness (also called True Positive Rate, TPR), and F-measure. After the classification, it is possible to obtain four types of results: True Positive (TP), True Negative (TN), False Positive (FP), and False Negative (FN). Let us explain the meaning of these terms: True Positive: Its value represents the number of instances that have been correctly classified as violent. False Negative: Its value represents the number of neutral videos that have been misclassified as violent programs. False positive: Its value represents the number of normal classes that have been misclassified as violent. True negative: Its value represents the number of normal classes that have been correctly classified as normal. Then, to estimate the accuracy, use the following definition and formula.

### Accuracy

Accuracy is the ratio of the number of correct predictions to the total number of test samples. It tells us whether a model is being trained correctly and how it may perform generally. Nevertheless, it works well if only each class has an equal number of elements.

$$\text{Accuracy} = \frac{\text{TP} + \text{TN}}{\text{TP} + \text{TN} + \text{FP} + \text{FN}}. \tag{1}$$

### Precision

Precision tells how often a prediction is correct when the model predicts positive. So precision measures the portion of positive identifications in a prediction set that were

**Table 7** Datasets for violence detection in video.

| Dataset | Reference | Characteristics | Published year | Reference used |
|---|---|---|---|---|
| Movies | *Bermejo et al. (2011)* | 200 video clips of totally 6 min | 2011 | *Serrano Gracia et al. (2015), Senst et al. (2017), Fenil et al. (2019), Sudhakaran & Lanz (2017), Ullah et al. (2019)* |
| Hockey | *Nievas et al. (2011)* | 1 000 video clips of totally 27 min | 2011 | *Serrano Gracia et al. (2015), Senst et al. (2017), Zhang et al. (2016), Gao et al. (2016), Deepak, Vignesh & Chandrakala (2020), Ding et al. (2014); Sudhakaran & Lanz (2017), Meng, Yuan & Li (2017), Serrano et al. (2018), Ullah et al. (2019)* |
| UCF-101 | *Soomro, Zamir & Shah (2012)* | 13 000 clips | 2012 | *Serrano Gracia et al. (2015)* |
| CASIA Action | *Wang, Huang & Tan (2007)* | 8 classes of single person activities that contain 1446 video clips | 2007 | *Yao et al. (2021)* |
| UT-Interaction | *Zhang et al. (2012a), Zhang et al. (2012b)* | 20 video sequences with the resolution of 720x480 at 30fps. | 2012 | *Yao et al. (2021)* |
| The Boss | *Bas, Filler & Pevný (2011)* | 10,000 images for training and 1,000 for testing | 2011 | *Arceda et al. (2016)* |
| XD-Violence | *Wu et al. (2020a), Wu et al. (2020b)* | 4574 videos with duration of 217 h that has 6 types of violent and 9 types of non-violent videos | 2020 | *Wu et al. (2020a), Wu et al. (2020b)* |
| VVAR10 | *Xie et al. (2016)* | 296 positive and 277 negative instances | 2016 | *Xie et al. (2016)* |
| UCF Sports | *Soomro & Zamir (2014)* | – | 2014 | *Xie et al. (2016)* |
| UCF50 | *Reddy & Shah (2013)* | 50 actions, 100 min videos | 2010 | *Xie et al. (2016)* |
| HMDB51 | *Kuehne et al. (2011)* | 6,766 manually annotated videos that divided to 51 classes | 2011 | *Xie et al. (2016)* |
| BEHAVE | *Blunsden & Fisher (2010)* | 200 000 frames | 2010 | *Zhang et al. (2016), Song, Kim & Park (2018), Serrano et al. (2018)* |
| CAVIAR | *Bins et al. (2005)* | 28 video clips | 2004 | *Zhang et al. (2016)* |
| Violent-Flow | *Xu et al. (2018)* | 30 video clips | 2012 | *Gao et al. (2016), Sudhakaran & Lanz (2017)* |
| UCF | *Sultani, Chen & Shah (2018)* | 128 h of videos. 1900 surveillance videos that includes 13 types of classes | 2018 | *Kamoona et al. (2019)* |
| Crowd Violence | *Song et al. (2019)* | – | 2012 | *Deepak, Vignesh & Chandrakala (2020), Meng, Yuan & Li (2017)* |

**Table 7** (*continued*)

| Dataset | Reference | Characteristics | Published year | Reference used |
|---|---|---|---|---|
| Finnish emotional | *Vaaras et al. (2021)* | 132 samples | 2021 | *Ye et al. (2021)* |
| Chinese emotional | *Poria et al. (2018)* | 370 samples | 2018 | *Ye et al. (2021)* |
| Pittsburgh | *Torii et al. (2013)* | – | 2013 | *Arandjelovic et al. (2016)* |
| Tokyo 24/7 | *Torii et al. (2015)* | – | 2015 | *Arandjelovic et al. (2016)* |
| Violent interaction | *Fenil et al. (2019)* | 2314 movies with 1077 fights and 1237 no-fights | 2019 | *Fenil et al. (2019)* |
| MediaEval | *Demarty et al. (2014)* | 10 000 clips | 2014 | *Mu, Cao & Jin (2016)* |
| Violent-Flows | *Rota et al. (2015)* | - | 2015 | *Sudhakaran & Lanz (2017)* |
| Weizmann | *Blank et al. (2005)* | 9 actions, 9 clips | 2005 | *Naik & Gopalakrishna (2021)* |
| KTH | *Schuldt, Laptev & Caputo (2004)* | 6 actions, 100 clips | 2004 | *Naik & Gopalakrishna (2021)* |
| Violent Crowd | *Bermejo et al. (2011)* | 246 short video sequences that video length is varying from 50 to 150 frames. | 2012 | *Ullah et al. (2019)* |
| London Metropolitan Police | Cheng et al. (2012) | – | 2012 | *Senst et al. (2017)* |

actually correct.

$$\text{precision} = \frac{TP}{TP + FP}.$$ (2)

## Recall

Recall is the number of correct positive results divided by the number of all relevant samples so recall represents the proportion of actual positives that were identified correctly.

$$Pr = \frac{TP}{TP + FN}.$$ (3)

## F1-score

F1-score is a measure of the test's accuracy It is the Harmonic Mean between precision and recall. The value of the F1 Score can be between 0 and 1. When the F1 score is equal to 1, the model is considered to work perfectly.

$$F_{\text{measure}} = \frac{2 \times \text{precision} \times \text{recall}}{\text{precision} + \text{recall}}.$$ (4)

## AUC-ROC

When predicting the probability, the greater we can get the true positive rate (TPR) at a lower false positive rate (FPR), the better the quality of the classifier. Therefore, we can introduce the following metric that evaluates the quality of the classifier that calculates the probability of an object belonging to a positive class:

$$AUC = \int_0^1 TPRdFPR.$$ (5)

This value is the area under the ROC curve. Here, AUC $\in [0, 1]$. The ROC curve is a graphical tool for evaluating the accuracy of binary classification models. It allows to find the optimal balance between sensitivity and specificity of the model, which corresponds to the point of the ROC curve closest to the coordinate (0,1), in which sensitivity and specificity are equal to 1, when there are no false-positive and false-negative classifications.

### False alarm

False Alarm: volume of false classes relative to the sum of classes.

$$FA = \frac{FP}{TN + FP}. \qquad (6)$$

### Missing alarm

Missing Alarm (MA): volume of false classes relative to the sum of classes.

$$MA = \frac{FN}{TP + FN}. \qquad (7)$$

## CHALLENGES TO VIOLENCE DETECTION IN VIDEO

Due to the many challenges faced when capturing moving people, detecting aggressive conduct is tough. The major problems that need to be addressed are discussed below.

### Dynamic illumination variations

When studying images with shifting light, which is a prevalent feature of realistic surroundings, tracking becomes problematic. When collecting video at night, outside CCTV cameras are exposed to environmental lighting variations, which might result in low contrast images, making content interpretation complicated. *Zhou et al. (2018)* has shown a better susceptibility to light fluctuation. The authors employed the LHOG descriptor, that is derived from a cluster of nodes. The LHOG features consequence from color space. In this research, two ways to deal with light variation were utilized: initially, the authors cut the block pitch in half, resulting in a half-block overlap. The normalization process is then carried out per LHOG. The motion magnitude photos are used to derive the LHOF, which catches real-time data. Furthermore, the adaptive background subtraction technique provides a dependable way for dealing with illumination variations, as well as repeated and long-term scenario alterations.

### Motion blur

This is a difficult challenge for optical flow-based motion estimates to solve: Parts of human body as head, arms, legs, elbows, and shoulders are mathematical feature points that create distinctive abstraction of different stances. *Deniz et al. (2014)* proposed an approach for monitoring high accelerations that does not require tracking. Extreme acceleration causes visual blur, making tracking less accurate or impossible, according to the researchers. Camera movement may, in fact, produce picture blur. They used a deconvolution primary

processing to get rid of it. To deduce global motion between each couple of successive frames, the phase of correlation approach is applied initially. If global motion is identified, the predicted slope and length of displacement are utilized to create a Point Spread Function and deconvolve next frame applying Lucy-Richardson iterative deconvolution technique.

## Presence of a non-stationary background

Since low-resolution movies often feature background movement caused by camera movement or changes in illuminance, noise correction is required. While the amplitude of the optical flow vector is a very powerful signal for detecting the degree of movement, and the flow direction may offer additional motion information, a recent study uses the optical flow technique for motion analysis (*Mahmoodi & Salajeghe, 2019*). The optical flow among each adjacent frames is computed using a background motion-resilient technique. The backdrop normally moves at a consistent pace in response to the camera movement. Human actions are more prone to move in irregular patterns. This enabled noise from background movement to be filtered. A 3*3 Gaussian kernel is used to minimize noise, a histogram equalization is used to disperse pixel intensities across a greater contrast range, and a background subtraction utilizing Mixture of Gaussians is used to exclude items not linked to the actors. The overwhelming background components may obstruct action recognition predictive performance. To address this problem, we must consider both localizing action samples and reducing the effect of backdrop video. Because of the camera movement, *Wang, Wang & Fan (2021)* noted a significant degree of horizontal movement in the backdrop. They proposed adding warped optical flow as an input modality, inspired by enhanced dense trajectory (*Bermejo et al., 2011*). By predicting the homography matrix and accounting for camera motion, they were able to extract them. That helps to concentrate on the performer by removing the background motion. The temporal segment network architecture receives additional aggregating capabilities in *Wang et al. (2018)*. This is a good way to emphasize crucial pieces while reducing ambient noise.

## Non-professionally produced content

The volume of non-professionally generated material has expanded dramatically in recent years. People can take images and videos everywhere and at any time owing to the widespread use of cameras and cellphones. Social media and online forums are used to disseminate the material. Only a little amount of study has been done to explore the scene structure of non-professionally generated material, despite the fact that journalists often use amateur content, particularly in news coverage on TV channels and news sites on the Internet. For example, if no professional team was presented at an occurrence or if one was submitted but missed a potentially interesting event. We see possibilities for a new discipline of social media-focused video scene identification algorithms. Films published on social networks sharing sites are typically brief, with added handheld camera movements, and the quality of two scenes might vary greatly. The new problem is not recognizing scenes in these short films, but rather identifying a scene (situation) that occurred in real life among a large number of videos on a social media sharing platform that incorporates footage from many sources and of varying quality. A scene may be displayed from several

perspectives, exposing more information, and so providing a better experience for the viewer, by mixing footage from different individuals.

## Few publicly available datasets

Because this is a relatively new study topic, there are much fewer publically accessible datasets for violence detection in video. Furthermore, the data imbalance between positive and negative samples precludes the implementation of supervised models. The lack of ground truth data as well as the ambiguous character of the anomalies makes it difficult to develop end-to-end trainable deep learning models (*Lloyd et al., 2016*). The modeling is further confounded by the large variation within positive instances (anomalous occurrences may include a wide range of distinct instances, despite the fact that most training data is restricted) (*Zhu et al., 2018*). As a result, there is a need for appropriate standards to assess the methods employed for violence detection in videos (*Constantin et al., 2020*).

## Computational and time-consuming cost

In general, the time-consuming phase of feature representation during video violence detection, which is both computationally and time-consuming, serves as a significant barrier for the implementation of violence detection in real applications (*Constantin et al., 2020*; *Popoola & Wang, 2012*). As a result, the majority of current algorithms for detecting violence in videos have significant time and space complexity costs. As a result, these techniques are unsuitable for practical uses (*Vu et al., 2020*). As a result, for improved feature extraction and description, more dense and deeper DNN models are required. In addition, simple, effortless, and more effective methods for detecting violence are required. However, the high dimensional structure coupled with a non-local change between frames increases the complexity of the algorithms used to identify video abnormalities.

## DISCUSSION

Despite considerable advancements in the video-based physical detection area, certain restrictions remain, making it more complex and demanding. In reality, selecting the features that make a moving item is a challenging task since it has a major impact on the behavior's description and analysis. It's problematic, for instance, to describe the action when the scene's backdrop changes often or when new items appear unexpectedly in the scene. Furthermore, the appearance of the moving object may be affected by a variety of variables, including clothes (dress, suit, footwear, *etc.*) and scene location (outdoor/indoor, *etc.*). In order to collect meaningful information regarding an object's behavior, features that are resistant to scene modifications (rotating, occlusion, blurring, cluttered backdrops, *etc.*) and less susceptible to changes in the object's appearance must be used. Furthermore, most algorithms for detecting violence assume that the moving item is located in front of the camera. In fact, though, the point of view is arbitrary. There are some studies that utilize several cameras to record various perspectives of the moving item and then unite them to circumvent this constraint. Even if such methods are efficient and provide excellent results, they are complex, time-consuming, and unsuitable for practical uses. On the other hand, depending on the context in which the action is done, as well as the time and location of

the action, the observed behavior may have many interpretations. Hugging, for example, is a common everyday activity for most people, but hitting, kicking, use fighting techniques are considered aberrant conduct that must be alarmed. On the other hand, depending on the context in which the action is done, as well as the time and location of the action, the observed behavior may have many interpretations. To address the constraints stated above, suggested methods utilized a massive quantity of training data that included all permissible situations. To cope with huge amounts of data, cloud computing has become popular, since it enables sophisticated algorithms, such as deep learning, to operate efficiently on larger datasets. In reality, because of their deep structures, the usage of deep learning techniques has exploded in order to achieve considerably more learning capacity.

Among the 80 reviewed papers, four articles (5%) are classified as review articles (*Shidik et al., 2019*; *Ramzan et al., 2019*; *Pawar & Attar, 2019*; *Sreenu & Durai, 2019*). The article reviews 220 papers that published between 2010 and 2019. It provides a review about methods, frameworks, techniques in video-based intelligent violence detection systems. While it asserts that its primary contribution is a thorough overview of the state-of-the-art of intelligence video surveillance, several significant shortcomings should be highlighted. Intelligence video surveillance was considered superficially without delving into the essence of the methods used. Moreover, the paper included only journal papers into their study that excludes state-of-the-art researches of high value conferences as ICCV, ECCV, ICML and CVPR. Thirdly, main accent directed to video surveillance as protection, object detection, activity recognition, traffic controlling, and disaster and accident monitoring that far from specific violence detection area.

*Ramzan et al. (2019)* explores violence detection techniques using machine learning methods. The review is not systematic as it was written in a free form. Authors classify video-based violence detection into three categories as machine learning based, SVM based and convolutional neural network based violence detection techniques. These strategies are discussed in depth, as well as their advantages and disadvantages. Additionally, thorough tables detail the datasets and video attributes that are employed in all procedures and play a critical part in the violence detection process.

This article aims at studying and analyzing deep learning techniques for video-based anomalous activity detection. As outcome of the study, the graphical taxonomy has been put forth based on kinds of anomalies, level of anomaly detection, and anomaly measurement for anomalous activity detection. The focus has been given on various anomaly detection frameworks having deep learning techniques as their core methodology. Deep learning approaches from both the perspectives of accuracy oriented anomaly detection and real-time processing oriented abnormality detection are compared. This study also sheds light upon research issues and challenges, application domains, benchmarked dataset and future directions in the domain of deep learning based anomaly detection (*Pawar & Attar, 2019*).

The next section of the evaluation delves further into object identification, action recognition, crowd analysis, and ultimately violence detection in a crowd setting (*Sreenu & Durai, 2019*). The article superficially examines research in the indicated areas without a deep disclosure of the methods used. The authors present four main approaches as

a classification of abnormal behavior: Hiden Markov Model, Gaussian Mixture Model (GMM), optical flow, and STT. Also, 13 datasets for the violence detection were shown. Nevertheless, as the purpose of the paper is related to different domains, the presented datasets also dedicated to different applications of deep learning in video surveillance. There are only four popular datasets in violence detection as CAVIAR, BEHAVE, Movie, and Hockey dataset. The rest are not directly related to the detection of violence.

Therefore, to the authors' knowledge, in the present literature, there is still a lack of a more formal and objective systematic review that is specifically focused on "Violence Detection Techniques in Video Surveillance Security Systems" and analyses it from multiple perspectives. The contributions that are made in this work, as mentioned in 'Introduction', can be used to address this issue.

## CONCLUSION

With the increasing development of surveillance cameras in many areas of life to watch human behavior, the need for systems that automatically identify violent occurrences increases proportionally. Violent action detection has become a prominent subject in computer vision attracting new researchers. Many academics have suggested various methods for detecting such actions in videos. The primary aim of this systematic review is to examine the most recent studies in the field of violence detection. The various types of video violence detection techniques, which perform using machine learning, SVM, and deep learning were examined in this study. First, we looked at the most common techniques for extracting and describing features. Furthermore, all datasets and video characteristics were utilized in all techniques, as well as those that play a critical part in the identification process are documented in thorough tables. The accuracy of object identification, feature extraction, and classification methods, as well as the dataset utilized, are all factors. Following that, we gave a thorough review of descriptors for violence detection. In addition, we discussed the most difficult datasets and assessment criteria for video violence detection methods. Finally, we discussed challenges, open issues, and future directions for violence detection in video. Our research may help to highlight the strategies and procedures for detecting violent behavior from surveillance videos.

### Funding

This research was supported by the grant "Development of artificial intelligence enabled software solution prototype for automatic detection of potential facts of physical bullying in educational institutions" funded by the Ministry of Education of the Republic of Kazakhstan. Grant No. IRN AP08855520. The funders had no role in study design, data collection and analysis, decision to publish, or preparation of the manuscript.

### Grant Disclosures

The following grant information was disclosed by the authors:
The Ministry of Education of the Republic of Kazakhstan: IRN AP08855520.

## Competing Interests

Batyrkhan Omarov, Sergazi Narynov, and Zhandos Zhumanov are employees of Alem Research.

## Author Contributions

- Batyrkhan Omarov conceived and designed the experiments, performed the experiments, analyzed the data, performed the computation work, prepared figures and/or tables, authored or reviewed drafts of the paper, and approved the final draft.
- Sergazi Narynov conceived and designed the experiments, performed the experiments, analyzed the data, performed the computation work, prepared figures and/or tables, and approved the final draft.
- Zhandos Zhumanov conceived and designed the experiments, analyzed the data, performed the computation work, prepared figures and/or tables, authored or reviewed drafts of the paper, and approved the final draft.
- Aidana Gumar and Mariyam Khassanova conceived and designed the experiments, performed the experiments, prepared figures and/or tables, authored or reviewed drafts of the paper, and approved the final draft.

## Data Availability

This is a literature review.

## Supplemental Information

Supplemental information for this article can be found online at http://dx.doi.org/10.7717/peerj-cs.920#supplemental-information.

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
