# Peer review of "State-of-the-art violence detection techniques in video surveillance security systems: a systematic review"

_PeerJ Computer Science, doi:10.7717/peerj-cs.920_

## Round 0.1 · original submission · Major Revisions

Please address the issues raised by the reviewers and prepare a revised manuscript.

The reviewers have suggested that you cite specific references. You are welcome to add it/them if you believe they are relevant. However, you are not required to include these citations, and if you do not include them, this will not influence my decision.

Reviewer 1 ·

Basic reporting

no comment

Experimental design

no comment

Validity of the findings

no comment

Additional comments

This paper provided a survey by the assessment of video violence detection problems that occurs in state-of-the-arts for both qualitative and quantitative studies in terms of procedure, datasets, and performance indicators. Some challenges and future directions are also covered. The topic seems to be very interesting, however, I have some major concerns about the paper that will further enhance the paper quality and its body structure.

The wording style of the abstract is too sloppy and instead of actual contents description, the background studies is largely covered. Authors need to make the abstract more compact and representative for the whole paper contents.

A reader gets confuse when face the sections without numbering that is an important aspect of paper body. Resolve this issue.

I did not find any novelty after going through the contribution lists that are already the practice of existing surveys such as methods coverage, their features extraction sets, datasets, etc. Authors are suggested to clearly highlight their contributions and mention why this survey is needed if there already exist several violence detection surveys.

I did not find the any visual statistical information of year-wise violence detection papers distribution. Authors need to include the details of paper coverage from each year, present their taxonomy, broadly categorize them into machine learning or conventional techniques and deep learning by investigating each year for the category. Similarly, authors are suggested to present visual or tabular representation of working flow of the survey for the ease of readers. Next, I did not most recent violence detection literature that authors need to includes such as:
An intelligent system for complex violence pattern analysis and detection. International Journal of Intelligent Systems. 2021 Jul 5
AI assisted Edge Vision for Violence Detection in IoT based Industrial Surveillance Networks. IEEE Transactions on Industrial Informatics. 2021 Sep 29
Violence detection and face recognition based on deep learning. Pattern Recognition Letters. 2021.

Challenges and future directions are not well-structured as they are explained in wordy manners. It is suggested to explain each challenge in separate small section.

Finally, I recommend to consider the English proficiency in terms of spelling and grammatical corrections.

·

Basic reporting

This systematic review provide a comprehensive assessment of the video violence detection problems that have been described in state-of-the-art researches. This paper deals with an interesting topic especially in context of video violence. Therefore, it requires revisions to improve the quality of work.

1. More suitable title should be selected for the article. Please use different terms in the "Title" and the "Keywords". The abstract should be ordered by answering questions such as Originality of the manuscript, Objectives, Method (indicate how many papers you located in the different stages of the SLR and how many you were finally left with) and Finding.

Experimental design

2. I suggest a new Table with SLR and bibliometric analysis have been used in ``several previous research papers''.
3. A flowchart should be added to the article to show the research methodology. In this sense, I propose to update the methodology as this paper does:
B. Kitchenham and S. Charters, "Guidelines for performing sandstematic literature reviews in software engineering version 2.3", Engineering, vol. 45, no. 5, pp. 1051, 2007.
In this paper you will see that the flowchart in figure 2 presents the Number of articles in each stage after applying the inclusion and exclusion criteria.
Abarca, V. M. G., Palos-Sanchez, P. R., & Rus-Arias, E. (2020). Working in virtual teams: a systematic literature review and a bibliometric analysis. IEEE Access, 8, 168923-168940.

Validity of the findings

4. Consider the length of the conclusions.
5. Add DOI for all references.
6. It is suggested to include a SLR protocol and question research.
7. It is suggested to compare the results of the present research with some similar systematic literature review which is done before.

Additional comments

The manuscript is of interest and examines a problem of great interest. However, the methodology applied needs to be improved.

---

## Round 0.2 · accepted · Accept

I congratulate the authors for making the revisions. I concur with the reviewers to accept the paper.

Reviewer 1 ·

Basic reporting

-

Experimental design

-

Validity of the findings

-

Additional comments

The authors have made significant improvements. I accept the paper.

I have added an optional minor comment.

Section 4 (CLASSIFICATION OF VIOLENCE DETECTION TECHNIQUES), the text details in the subsections are too long. Please add few headings in each subsection for readers understanding.